# The Study of Thermal Stability of Mn-Zr-Ce, Mn-Ce and Mn-Zr Oxide Catalysts for CO Oxidation

**DOI:** 10.3390/ma15217553

**Published:** 2022-10-27

**Authors:** T. N. Afonasenko, D. V. Glyzdova, V. L. Yurpalov, V. P. Konovalova, V. A. Rogov, E. Yu. Gerasimov, O. A. Bulavchenko

**Affiliations:** 1Center of New Chemical Technologies BIC, Boreskov Institute of Catalysis, Neftezavodskaya st., 54, Omsk 644040, Russia; 2Boreskov Institute of Catalysis SB RAS, Lavrentiev Ave. 5, Novosibirsk 630090, Russia

**Keywords:** Mn-Ce, Mn-Zr, Mn-Zr-Ce, thermal stability, CO oxidation

## Abstract

MnO_x_-CeO_2_, MnO_x_-ZrO_2_, MnO_x_-ZrO_2_-CeO_2_ oxides with the Mn/(Zr + Ce + Mn) molar ratio of 0.3 were synthesized by coprecipitation method followed by calcination in the temperature range of 400–800 °C and characterized by XRD, N_2_ adsorption, TPR, TEM, and EPR. The catalytic activity was tested in the CO oxidation reaction. It was found that MnO_x_-CeO_2_, MnO_x_-ZrO_2_-CeO_2_, MnO_x_-ZrO_2_ catalysts, calcined at 400–500 °C, 650–700 °C and 500–650 °C, respectively, show the highest catalytic activity in the reaction of CO oxidation. According to XRD and TEM results, thermal stability of catalysts is determined by the temperature of decomposition of the solid solution Mn_x_(Ce,Zr)_1−x_O_2_. The TPR-H_2_ and EPR methods showed that the high activity in CO oxidation correlates with the content of easily reduced fine MnO_x_ particles in the samples and the presence of paramagnetic defects in the form of oxygen vacancies. The maximum activity for each series of catalysts is associated with the start of solid solution decomposition. Formation of active phase shifts to the high-temperature region with the addition of zirconium to the MnO_x_-CeO_2_ catalyst.

## 1. Introduction

Catalytic CO oxidation is one of the most effective techniques for removing this pollutant, which is formed during the incomplete combustion of hydrocarbon fuels [1]. Materials based on noble metals are often used as a catalyst for CO oxidation, but the price of such samples and restricted availability of the noble metals makes their use economically unreasonable, helping to prevent their wider application. Thus, development of more economical, cheaper and affordable catalysts is of great importance [1,2]. An example of such non-noble systems can be manganese-containing compounds in the form of pristine or mixed oxides, which are widely studied as effective catalysts for the CO oxidation [3,4], as well as systems for soot oxidation [5], combustion of volatile organic compounds [6,7,8,9,10] and NO_x_ removal [11,12,13]. The catalytic properties of Mn-containing systems are due to the ability of manganese to exist in the form of various stoichiometric (MnO_2_, Mn_5_O_8_, Mn_2_O_3_, Mn_3_O_4_, MnO and their polymorphs) and nonstoichiometric oxides, which are characterized by a variable oxidation state of manganese between +4 and +2 [14].

The calcination temperature is an important parameter that determines the properties of manganese catalysts. In the case of manganese oxides, it has a significant effect on the surface area of the samples, the states of manganese ions, and ethe nature and content of reactive oxygen species, including adsorbed oxygen, oxygen vacancies, and mobile oxygen. These factors determine the catalytic activity of MnO_x_ [15]. The effects of high-temperature treatment on mixed manganese-containing oxides can lead to phase transformations associated with the formation and subsequent decomposition of joint phases, which should also affect the catalytic activity. Thus, a significant increase in activity was found for MnO_x_-Al_2_O_3_ catalysts calcined at 950–1000 °C in the reaction of CO and hydrocarbons (butane, benzene, and cumene) oxidation [16]. It was established earlier [17] that the Mn_3−x_Al_x_O_4_ (x~1.5) mixed oxide is formed during high-temperature treatment. Subsequent cooling in air leads to the addition of oxygen to the Mn_3−x_Al_x_O_4_ phase and its partial decomposition with the formation of highly dispersed β-Mn_3_O_4_ particles, which increases the amount of weakly bound oxygen, causing an increase in catalytic activity.

MnO_x_-CeO_2_ are the most catalytically active manganese-containing oxides [18]. The incorporation of manganese cations into the ceria lattice with the formation of solid solution causes a decrease in crystallinity, creating more lattice defects and oxygen vacancies, which significantly improves the ability of CeO_2_ to accumulate oxygen, and also increases the mobility of oxygen on its surface [19]. However, MnO_x_-CeO_2_ has low thermal stability. Calcination above 500 °C leads to its deactivation, due to the decomposition of solid solution and a sharp decrease in the surface area [7,20,21]. One of the ways to improve the thermal stability of MnO_x_-CeO_2_ is by the addition of zirconium, which inhibits the agglomeration of oxide particles, forming a joint phase with CeO_2_ [22,23,24]. Zirconium also has a similar effect on manganese oxide [25]. According to Zeng et al. [26], the significant activity improvement in propane oxidation after zirconium addition to MnO_x_ (after calcination at 500 °C) was attributed to the formation of solid solution, the superior redox ability, a higher oxygen mobility and a more abundant oxygen vacancy. Previously, we found [27] that Mn_0.12_Zr_0.88_O_x_ samples calcined at 650–700 °C exhibit the highest catalytic activity in CO oxidation compared to catalysts calcined at other temperatures, which is explained by the partial decomposition of solid solution and formation of highly dispersed MnO_x_ particles on its surface.

Based on the properties of MnO_x_-CeO_2_ and MnO_x_-ZrO_2_, it can be assumed that the MnO_x_-ZrO_2_-CeO_2_ oxide should combine high activity and thermal stability. Long et al. [28] showed that MnO_x_-ZrO_2_-CeO_2_ oxides, compared to MnO_x_-ZrO_2_ and MnO_x_-CeO_2_ samples calcined at the same temperature of 500 °C, have the highest catalytic activity in the oxidation of chlorobenzene due to the high content of Mn4+ and surface oxygen, and higher mobility of oxygen in the solid solution. A similar comparison of supported catalysts calcined at 500 °C in the process of NO removal [29] showed the advantage of MnO_x_/CeO_2_ and MnO_x_/CeO_2_-ZrO_2_ compared to MnO_x_/ZrO_2;_ according to the authors, this is associated with a higher dispersion of manganese oxide and a higher content of weakly bound oxygen in MnO_x_/CeO_2_ and MnO_x_/CeO_2_-ZrO_2_ catalysts. These comparative studies demonstrate the high activity of the MnO_x_-ZrO_2_-CeO_2_ system; however, the calcination temperature of such catalysts usually does not exceed 550 °C [30,31]. Moreover, information about the thermal stability of MnO_x_-ZrO_2_-CeO_2_ is limited to the studies of its properties at two or three calcination temperatures (500–550 °C, 800–900 °C). Such studies are carried out to estimate the resistance of the sample to possible overheating [24] or to simulate the "aging" of the catalyst [22]. To the best of our knowledge, no such studies of changes in the structural and catalytic properties of MnO_x_-ZrO_2_-CeO_2_ under the influence of calcination temperature in comparison with the properties of MnO_x_-CeO_2_ and MnO_x_-ZrO_2_ oxides have been reported. These kinds of studies are important in developing a fundamental understanding of the thermal stability of the Mn-containing catalyst.

In current work we study the effect of a calcination temperature in the range of 400–800 °C on the catalytic activity and structural properties of MnO_x_-ZrO_2_-CeO_2_, MnO_x_-CeO_2_, and MnO_x_-ZrO_2_ catalysts for CO oxidation. Comparison of three series of catalysts would help to determine the structure—activity relationship, including the role of the cationic composition in formation of solid solution, the temperature of decomposition, and the change in the catalytically active states.

## 2. Materials and Methods

### 2.1. Catalytic Synthesis

The samples were prepared by the precipitation method. A joint solution of ZrO(NO_3_)_2_, Ce(NO_3_)_3_, and Mn(NO_3_)_2_ salts was obtained, and the NH_4_OH solution was gradually added into the mixture with continual stirring until the pH reached 10. The precipitation process was carried out at 80 °C. Stirring of the suspension was continued for 1 hour, then H_2_O_2_ was added dropwise to ensure the completeness of precipitation. The amount of H_2_O_2_ corresponded to the molar ratio of H_2_O_2_:(Mn + Zr + Ce). The suspension was finally kept without stirring for 2 hours. The obtained precipitate was filtered off and washed with distilled water on the filter to pH = 6–7. The samples were dried at 120 °C (2 h) and calcined in a muffle furnace at 400–800 °C for 4 h. The molar ratio of Mn:Zr:Ce in the prepared samples was 0.3:0.35:0.35.

Mn-Zr and Mn-Ce binary samples with the molar ratio Mn:Zr(Ce) 0.3:0.7 were prepared by a similar way.

The samples were designated as Mn-Zr-Ce-T, Mn-Ce-T and Mn-Zr-T, where T is the calcination temperature (in °C).

### 2.2. Catalytic Characterization

The specific surface area of the catalysts (S_BET_) was determined by the Brunauer–Emmett–Teller method using nitrogen adsorption isotherms measured at liquid nitrogen temperature. The studies were carried out using an ASAP 2400 automated system (Micromeritics Instrument Corp., Norcross, GA, USA).

X-ray diffraction analysis (XRD) of Mn-Zr, Mn-Ce and Mn-Zr-Ce catalysts was performed on a Bruker D8 Advance diffractometer using Cu-Kα radiation (λ = 1.5418 Å) in the 2θ angle range of 15–90° at scan step 0.05°. The accumulation time was 4 s.

Transmission electron microscope (TEM) analysis was performed by ThemisZ electron microscope (Thermo Fisher Scientific, Waltham, MA, USA) equipped with a corrector of spherical aberrations, with the accelerating voltage of 200 kV. Images were recorded using Ceta 16 CCD sensor (Thermo Fisher Scientific, Waltham, MA, USA). Elemental maps were obtained using energy dispersive spectrometer SuperX (Thermo Fisher Scientific, Waltham, MA, USA).

Thermally programmed reduction (TPR-H_2_) was carried out in a quartz reactor using a flow unit equipped with a thermal conductivity detector. A gas mixture (10 vol.% H_2_ in Ar) was fed to the reactor at a rate of 40 mL/min. The heating rate was 10 °C/min from room temperature to 900 °C.

Electron paramagnetic resonance (EPR) spectroscopy was performed at 25 °C on X-band (~9.7 GHz) EMXplus spectrometer (Bruker, Karlsruhe, Germany) with an ER 4105 DR resonator at a microwave power of 2.0 mW, a modulation frequency of 100 kHz and an amplitude of modulation of 1.0 G. The spectrometer was controlled by a personal computer and the Bruker WinEPR Acquisition program. The results were processed using the Bruker WinEPR Processing software. The EPR spectra were simulated by means of the Bruker WinEPR SimFonia (v.1.2). The spectra were recorded for ~20 mg of each catalyst placed in a quartz tube with an outer diameter of 5 mm.

Catalytic tests of the samples in the CO oxidation reaction were carried out in a glass flow reactor (170 mm × Ø 10 mm). The initial gas mixture included 1 vol.% of CO and 99 vol.% of air. The flow rate of the gas mixture through the reactor during the catalytic tests was changed in the range of 253–487 mL/min to vary the contact time. The reaction temperature was 150 °C. The CO conversion for each sample was determined at 3 different contact times (τ). A catalyst fraction of 0.4–0.8 mm was used. The catalyst weight varied from 0.2 to 1.5 g. To avoid overheating during the exothermic reaction of CO oxidation, the catalyst was mixed with quartz granules of the same fractional composition. The temperature in the catalyst bed was controlled and regulated using a chromel-alumel thermocouple connected to a heat controller.

The reaction mixture was analyzed by gas chromatography. The mixture was separated on a packed column filled with CaA zeolite (3 m). The unreacted amount of CO was determined using a thermal conductivity detector. The reaction rate (activity) R (cm^3^(CO)/(g*s)) was calculated from the CO conversion determined at different flow rates of the reaction mixture and taking into account the weight of the catalyst:R = [C_o_ − C] × V/(60 × m_cat_),
where C = C_o_ × (1 − ((P_o_ − P/P_o_)), m_cat_ is the catalyst weight (g), V is the flow rate of the reaction mixture (mL/min), C_o_ is the initial CO concentration in the gas mixture (vol.%), C is the CO concentration in the outlet flow (vol.%), P_o_ is the peak area on the chromatogram corresponding to the CO concentration in the initial reaction mixture, and P is the peak area on the chromatogram corresponding to the CO concentration in the outlet flow.

## 3. Results

### 3.1. Catalytic Properties

Figure 1 shows data on the catalytic activity of Mn-Ce, Mn-Zr and Mn-Zr-Ce samples calcined at 400–800 °C in the CO oxidation reaction. It can be noted that the form of the observed dependence is individual for each series of samples.

In the case of Mn-Ce system, the highest value of catalytic activity is typical for catalysts calcined at 400–500 °C and is 5.4–5.8 × 10^−2^ cm^3^(CO)/g*s. A further increase in the calcination temperature causes a sharp decrease in catalytic activity: the activity decreases to 3.2 × 10^−2^, 1.3 × 10^−2^ and 0.3 × 10^−2^ cm^3^(CO)/g*s for Mn-Ce samples calcined at 600, 700 and 800 °C, respectively.

The effect of the calcination temperature of Mn-Zr catalysts on their activity in the CO oxidation reaction differs from that observed for the Mn-Ce system. The catalytic activity of Mn-Zr catalysts calcined at 400–500 °C remains approximately at a constant level of 2.0–2.1 × 10^−2^ cm^3^(CO)/g*s. Raising the calcination temperature to 600 °C leads to an increase in activity up to 3.4 × 10^−2^ cm^3^(CO)/g*s. A further increase in the calcination temperature to 650 °C causes an almost twofold increase in the catalytic activity, so that the parameter reaches the value of 6.3 × 10^−2^ cm^3^(CO)/g*s. At the same time, the activity of the catalysts calcined at 700 °C slightly decreases to 5.5 × 10^−2^ cm^3^(CO)/g*s. A higher temperature treatment at 800 °C causes a sharp loss of activity to a value of 2.1 × 10^−2^ cm^3^(CO)/g*s, which coincides with the activity of the samples treated at 400 and 500 °C. Thus, the dependence of the catalytic activity of Mn-Zr catalysts on the calcination temperature has the form of an extremum; the highest activity is achieved for samples calcined at 650–700 °C.

In the case of Mn-Zr-Ce catalysts, an increase in the calcination temperature from 400 to 500 °C leads to an increase in activity from 3.0 to 5.4 × 10^−2^ cm^3^(CO)/g*s. A further increase in the treatment temperature up to 650 °C has no significant effect on the catalytic activity, while the Mn-Zr-Ce catalysts treated at 700 °C lose their activity, so that R does not exceed 4.4 × 10^−2^ cm^3^(CO)/g*s. Calcination at 800 °C leads to a further sharp drop in activity to 0.1 × 10^−2^ cm^3^(CO)/g*s. Thus, the temperature range required for the formation of the most catalytically active states in Mn-Zr-Ce samples is 500–650 °C.

Comparison of the kinetic data obtained for Mn-Ce, Mn-Zr and Mn-Zr-Ce oxides indicates that each of the studied systems is characterized by the calcination temperature range, where its samples demonstrate the highest catalytic activity relative to the samples of the other two systems. Such an optimal calcination temperature range for Mn-Ce catalysts is 400–500 °C. Mn-Zr catalysts are preferably calcined at 650–700 °C, while Mn-Zr-Ce triple oxide catalysts are in an intermediate position between Mn-Zr and Mn-Ce systems and exhibit high catalytic activity only in the calcination temperature range of 500–650 °C.

In order to explain the observed changes in the catalytic activity of Mn-Ce, Mn-Zr and Mn-Zr-Ce oxides depending on the calcination temperature, the changes in the structural properties of these systems were considered and compared.

### 3.2. Structural and Microstructural Properties

Figure 2a shows X-ray diffraction patterns of Mn-Ce catalysts. Peaks with maxima at 2θ = 28.6, 33.1, 47.5, 56.5, 59.2, 69.5, 76.9, 79.10, and 88.6°, corresponding to 111, 002, 022, 113, 222, 004, 133, 024, and 224 reflections of CeO_2_ fluorite structure (space group Fm-3m, PDF № 431002), are observed in the XRD patterns of all samples. An increase in the calcination temperature leads to a narrowing of the diffraction peaks and their shift towards smaller angles. As the temperature increases to 800 °C, additional peaks appear at 2θ = 23.2, 38.5, 55.2 and 65.9°, corresponding to 112, 004, 444, and 226 reflections of Mn_2_O_3_ (space group Ia-3, PDF № 41-1442), and 2θ = 36.1, corresponding to the most intense 121 reflection of Mn_3_O_4_ oxide (space group I4_1_/amd, PDF № 41-1442).

Diffraction patterns of Mn-Zr-Ce catalysts are shown in Figure 2b. It can be seen that catalysts calcined at 400–700 °C have similar X-ray patterns: broad peaks with maxima at 2θ = 28.9, 33.5, 48.2, 57.2, 60.0, 70.5, 78.0, 80.4, and 89.3°, corresponding to 111, 002, 022, 113, 222, 004, 133, 024, and 224 reflections of CeO_2_ fluorite structure, are observed. A low-intensity Mn_3_O_4_ reflection appears at 2θ = 36.3°, and its intensity increases with an increase in the calcination temperature. Significant changes occur in the diffraction patterns of Mn-Zr-Ce-800: the reflections of Mn_3_O_4_ appear and the fluorite reflections split. The latter may be related to decomposition of the initial oxide.

Figure 2c contains X-ray diffraction patterns of the Mn-Zr catalysts. The calcination of the samples at 400–500 °C leads to the appearance of a wide halo at 2θ = 25–40°, which indicates the formation of an X-ray amorphous state. The amorphous phase partially remains and induces an appearance of a background in the diffraction pattern of the sample calcined at 600 °C. The diffraction pattern of this sample also contains peaks at 2θ = 30.5, 35.4, 50.9, 60.6, 63.6, 74.9, 83.0, and 85.7°, which correspond to the ZrO2 oxide. Since the broadening of diffraction peaks, from XRD data it is not possible to correctly distinguish the cubic and tetragonal modification of zirconia. Diffraction signals of Mn_2_O_3_ oxide also appear at 2θ = 32.9, 38.2, 55.1, and 65.7°, corresponding to 222, 004, 444 and 226 reflections. These phases are present in the XRD patterns of Mn-Zr-650 and Mn-Zr-700 samples. High-temperature treatment at 800 °C causes the appearance of additional peaks at 2θ = 30.2, 34.6, 35.8, 50.2, 50.7, 59.3, 60.2, 62.9, 72.9, and 74.6°, corresponding to 011, 002, 110, 112, 020, 013, 121, 022, 004, and 220 reflections of the tetragonal modification of t-ZrO_2_ oxide (space group P4_2_/nmc, PDF № 54-1089). The monoclinic modification of m-ZrO_2_ also forms, as evidenced by the presence of a peak at 2θ = 28.2°, corresponding to the 111 reflection (space group P 2_1_/c, PDF № 37-1484). 

The structural characteristics of the catalysts calculated by the Rietveld method are given in Table 1. For Mn-Zr-600 and Mn-Zr-700, the observed lattice parameter of ZrO_2_ catalysts is 5.059–5.061 Å, which is much less than the literature parameter for pristine zirconium oxide (5.110 Å). The difference in the lattice parameters may indicate the formation of the Mn_y_Zr_1−y_O_2−δ_ solid solution, since the ionic radii of Zr^4+^ and Mn^3+^ are 0.84 Å and 0.66 Å, respectively [32]. The lattice parameter increases with an increase in the calcination temperature and becomes close to the value characteristic of pristine ZrO_2_ at 800 °C. Simultaneously, an increase in the content of the Mn_2_O_3_ phase to 9 wt.% is observed. These three factors indicate the decomposition of the Mn_y_Zr_1−y_O_2−δ_ solid solution, accompanied by the diffusion of Mn ions from the volume of oxide and the Mn_2_O_3_ phase formation: a decrease in the manganese content in the mixed oxide, a decrease in symmetry of zirconia, and the extrication of m-ZrO_2_. 

Similar trends have been established for the Mn-Ce and Mn-Zr-Ce series. In the case of Mn-Ce-400, the lattice parameter of the oxide is 5.360(1) Å, which is noticeably lower than that of pristine cerium oxide CeO_2_ (PDF № 431002, a = 5.411 Å) and indicates the formation of Mn_y_Ce_1−y_O_2−δ_ solid solution. An increase in the synthesis temperature causes an increase in the oxide lattice parameter to 5.406 Å. The gradual increase in the lattice constant to the value of pristine CeO_2_ phase indicates the diffusion of manganese cations from the solid solution structure to its surface. At 800 °C, Mn cations form crystalline manganese oxides Mn_2_O_3_ and Mn_3_O_4_, which are detected by XRD. In the case of the Mn-Zr-Ce series, the lattice parameters vary from 5.182(1) to 5.371(1) Å, indicating the formation of a Mn_y_Zr_x_Ce_1−y_O_2−δ_ solid solution. An increase in the calcination temperature from 400 to 600 °C leads to a decrease in the lattice constant from 5.337(1) to 5.316(1) Å and a simultaneous increase in the content of manganese oxide. The observed effect is associated with the decomposition of solid solution by diffusion of Mn cations from the initial oxide and the appearance of a “new” Mn_y2_Zr_x2_Ce_1−y2_O_2−δ2_ solid solution. Catalysts treated at 700–800 °C contain phases with lattice parameters of 5.345(1)–5.371(1) and 5.182(3)–5.214(1) Å, which tend to the values for pristine CeO_2_ and ZrO_2_ oxides. In this case, two mixed oxides are formed; the first one is based on ceria and the second one is based on zirconia. In addition, some heterogeneity in composition cannot be excluded for catalysts synthesized at lower temperatures. The average sizes of coherent scattering regions (CSR) grow with an increase in the calcination temperature from 50–110 Å at 400–600 °C to 130–240 Å at 800 °C. The introduction of Zr reduces the average CSR size of the solid solution formed at 700–800 °C as compared to the Mn-Ce catalysts.

Thus, it can be concluded that all three series are characterized by similar changes in the phase composition, depending on the synthesis temperature: an increase in temperature leads to decomposition of Mn_y_Zr_x_Ce_1−y_O_2−δ_ solid solutions with the formation of Mn_2_O_3_ and Mn_3_O_4_ crystalline oxides and cerium/zirconium oxides. Based on the lattice parameters’ change, it can be assumed that there is a temperature range, one lower than the temperature of crystalline manganese oxides formation, at which the decomposition of the initial solid solutions has already begun and manganese cations are on the oxide surface in the form of amorphous MnO_x_ particles (Table 1, Figure 2). It should be noted that an increase in the calcination temperature leads to a sharper change in the lattice parameters of the solid solutions formed in the Mn-Ce-Zr and Mn-Zr samples (at 700 °C) compared to the solid solution existing in the Mn-Ce system, which is characterized by a linear change in the lattice constant.

As expected, the specific surface area of the catalysts decreases with increasing the calcination temperature for all series (Table 1). The S_BET_ of X-ray amorphous Mn-Zr samples obtained at 400–500 °C is 304–306 m^2^×g^−1^. A further increase in the calcination temperature to 600 °C causes phase crystallization and a decrease in specific surface area to 176 m^2^×g^−1^. The decomposition of a solid solution at 800 °C leads to a decrease in S_BET_ down to 24 m^2^×g^−1^. The dependence of the specific surface area on the synthesis temperature of Mn-Ce catalysts is close to linear: the S_BET_ value is 72 m^2^×g^−1^ for the sample calcined at 400 °C and decreases to 9 m^2^×g^−1^ for the catalyst calcined at 800 °C. The specific surface area of the Mn-Zr-Ce triple oxide system calcined at 400 °C is 146 m^2^×g^−1^. Further, it decreases linearly with an increase in the calcination temperature within the temperature range of a solid solution existence up to 700 °C (63 m^2^×g^−1^). Raising the temperature to 800 °C leads to a sharp drop in S_BET_ to 12 m^2^×g^−1^. Accordingly, the observed changes in the specific surface area correlate with the phase transformations occurring in mixed oxides during an increase in the calcination temperature, and the thermal stability of each of the studied oxides is determined by the decomposition temperature of the corresponding solid solution.

The most active catalysts in each series (Mn-Ce-500, Mn-Zr-Ce-600, Mn-Zr-700) were studied by high-resolution transmission electron microscopy coupled with EDS mapping (Figure 3a–c). EDS mapping shows that for all cases there are areas enriched by Mn cations and a mixed Mn-Ce-Zr region. The latter is associated with formation of solid solution. In the case of Mn-Ce-500 and Mn-Zr-700 catalysts, the content of Mn (x) in mixed oxide Mn_x_(Ce,Zr)_1−x_O_2_ is 0.25–0.35. For Mn-Zr-Ce-600, several Mn-Zr-Ce regions can be distinguished in the region of the solid solution enriched in Ce and Zr cations. These results correlate well with XRD analysis (Table 1). In addition to areas containing three cations, one can distinguish regions enriched in manganese. Figure 3d–f illustrates the formation of various manganese oxides: Mn_3_O_4_, Mn_2_O_3_ and Mn_5_O_8_ for Mn-Ce-500, Mn-Zr-700, and Mn-Zr-Ce-600 catalysts, respectively. For Mn-Zr-Ce-600 catalyst, Mn_5_O_8_ was not detected by XRD (Table 1); the Mn oxide is probably in an XRD amorphous state. The binary catalysts are characterized by the presence of manganese oxides in the form of nanorods with particle sizes of 10–30 nm in diameter and 100–200 nm in length. For a ternary Mn-Zr-Ce-600 catalyst, manganese oxide has a rounded particle shape 20–30 nm in size.

### 3.3. Temperature-Programmed Reduction

Mn-Ce, Mn-Zr, and Mn-Zr-Ce catalysts synthesized at 500, 600 and 700 °C were studied by TPR-H_2_ in order to evaluate the differences in their redox properties. The choice of these calcination temperatures is due to the fact that each of the temperatures is optimal for achieving the highest catalytic activity for one of the three studied systems. The TPR-H_2_ profiles are shown in Figure 4.

The TPR curves of Mn-Ce catalysts show an intensive signal at high temperatures of ~600–900 °C with a maximum at 818 °C, which is explained by the reduction of Ce^4+^ ions in ceria [33,34,35,36,37]. In the range from 100 to 500 °C, the TPR profiles contain a broad signal of partially unseparated peaks with the main maxima at ~260–270 °C and 350–360 °C. These signals are associated with the consumption of hydrogen due to the sequential partial reduction of manganese cations present in the CeO_2_-based solid solution. The appearance of a signal at ~260–270 °C corresponds to the reduction of Mn^4+^/Mn^3+^ ions to Mn^3+^/Mn^2+^, while the peak at ~350–360 °C is due to the Mn^3+^/Mn^2+^ → Mn^2+^ transition [34]. In the case of Mn-Ce catalysts calcined at 500 and 600 °C, the first signal with a maximum at 260–270 °C has a shoulder at low temperatures (~220 °C). The consumption of hydrogen in this region can be explained by the reduction of finely dispersed MnO_x_ particles, which are probably formed on the surface of the solid solution during its decomposition. It should be noted that this signal does not appear on the TPR curve of the sample calcined at 700 °C, which may be due to the formation of Mn_3_O_4_ crystalline phase. The TPR profiles of Mn-Ce samples calcined at 600 and 700 °C also contain an additional low-intensity signal at ~500–520 °C, which can be explained by the reduction of Ce^4+^ ions located on the ceria surface or in the subsurface layer of the samples [33,34].

The TPR-H_2_ curves of Mn-Zr series are characterized by the presence of a complex, wide, intense signal in the reduction temperature range from ~100 to 500 °C, one associated with the occurrence of parallel processes of manganese reduction. Manganese can exist both in the composition of a solid solution based on ZrO_2_ and in nonstoichiometric highly-dispersive MnO_x_ particles [38,39], which can be formed during the decomposition of a solid solution at T ≥ 500 °C [40]. Similar to the Mn-Ce catalysts, sequential reduction processes of manganese ions Mn^4+^/Mn^3+^ → Mn^3+^/Mn^2+^ → Mn^2+^ also occur in the system [39]. It is most likely that the intense consumption of hydrogen at ~320 °C is a consequence of the Mn^3+^/Mn^2+^ phase transformation. It should be noted that the TPR-H_2_ curves of all Mn-Zr samples contain a low-intensity signal at ~550 °C, which can be associated with the partial reduction of zirconium oxide [40,41].

The reduction curves of Mn-Zr-Ce triple systems are shown in Figure 4c. A characteristic feature of the TPR profiles of Mn-Zr-Ce catalysts is the appearance of a broad low-intensity signal of the partial reduction of Ce^4+^ in the temperature range of 600–900 °C (T_max_~770–810 °C) [33,34]. The TPR profiles of the Mn-Zr-Ce systems, similarly to the TPR curves recorded for the Mn-Ce and Mn-Zr binary systems, also contain a wide signal of partially unseparated peaks in the temperature range of 100–500 °C, which are associated with the sequential phase transformations of manganese oxides: MnO_2_/Mn_2_O_3_ → Mn_3_O_4_ (the maximum at 260–270 °C) and Mn_3_O_4_ → MnO (the maximum at 360–370 °C) [42,43], as well as a gradual change in the oxidation state of manganese cations in the Mn_y_Zr_x_Ce_1−y_O_2−δ_ structure of solid solution. All Mn-Zr-Ce samples are also characterized by hydrogen consumption at low temperatures of 150–300 °C, which is expressed in the appearance of a shoulder on the TPR curves. We assume that this feature is explained by the reduction of highly dispersed MnO_x_ particles, which are formed during the decomposition of the Mn_y_Zr_x_Ce_1−y_O_2−δ_ solid solution and exist in a highly dispersed state on its surface. An increase in the calcination temperature of Mn-Zr-Ce catalysts to 600 °C leads to the appearance of an additional signal of hydrogen absorption with a maximum at ~430 °C. This signal may appear due to the peaks overlap for the reduction of manganese and Ce^4+^ ions located in the surface layer of the catalyst [44].

By fitting the TPR profiles with individual components, it was possible to calculate the fraction of the low-temperature signal at T~200 °C for each series of the samples (Table 2). In the case of Mn-Ce-500, the content of weakly bound oxygen is ~27% toward the total hydrogen uptake. Its amount decreases to ~18% with an increase in the calcination temperature to 600 °C, and a further increase in the calcination temperature causes the complete disappearance of the low-temperature signal. In the case of Mn-Zr catalysts, an increase in the synthesis temperature from 500 to 600 °C leads to an increase in the amount of weakly bound oxygen in the system from ~17 to ~29%, which can occur due to an increase in the amount of fine MnO_x_ particles. The total content of low-temperature signals observed in the TPR profile of Mn-Zr-700 is ~37%. An analysis of the TPR curves of the Mn-Zr-Ce triple oxide catalysts showed that the content of weakly bound oxygen in the samples calcined at 500 and 600 °C is ~12%, which corresponds to the states of solid solutions before their decomposition and may indicate the formation of dispersed amorphous MnO_x_ particles. A further increase in the calcination temperature to 700 °C leads to a decrease in the amount of the low-temperature signal to ~4%.

### 3.4. Electron Paramagnetic Resonance Spectroscopy

EPR spectra obtained for Mn-Ce, Mn-Zr and Mn-Zr-Ce samples (Figure 5) are similar for each series, but differ depending on the chemical composition of the catalyst. Mn-Ce and Mn-Zr-Ce samples have signals of sextet in EPR spectra, characteristic for Mn^2+^ cations [22,33], while in Mn-Zr samples there are no signals with a width less than 500 G, including Zr^3+^ [45] from the monoclinic phase of zirconium oxide.

A more thorough analysis and simulation of the EPR spectra made it possible to establish detailed differences in the spectra for the samples of the same composition activated at different temperatures. In particular, it was able to identify the components of the spectra associated with the signals of Mn^2+^, Mn^4+^ ions and paramagnetic defects (oxygen vacancies), as well as to evaluate their contribution to the overall spectrum. Mn^3+^ cations are EPR-silent under the conditions of the current experiment. It is necessary to use high-frequency EPR spectrometers for their registration [46].

The experimental and simulated EPR spectra of Mn-Ce catalysts are shown in Figure 6. The spectra are represented as a superposition of three types of signals with the following calculated parameters: g = 2.08–2.10, ∆H_pp_ = 600 G (paramagnetic defects/oxygen vacancies [45,47]); g = 1.998, ∆H_pp_ = 2000 G (Mn^4+^ cations [48]), as well as two Mn^2+^ cation signals [22,33] with g = 2.00, A = 92 G and different line widths of ∆H_pp_ = 240 G and ∆H_pp_ = 80 G (Mn-Ce-500), ∆H_pp_ = 65 G (Mn-Ce-600), ∆H_pp_ = 55 G. The narrowing of Mn^2+^ EPR-line in the spectra is usually associated with an increase in the interaction between Mn and Ce cations, namely, with a more active incorporation of manganese ions into the structure of cerium oxide [29]. The narrowing of Mn^2+^ signal lines with an increase in the calcination temperature for the Mn-Ce series can also be explained by a decrease in the dipole-dipole interaction of the nearest Mn^2+^ cations as a result of a decrease in their concentration in the two-component phase due to the diffusion of paramagnetic cations from the structure of the solid solution [49] and/or their partial transition to the EPR-inactive Mn^3+^ form. This phenomenon is also confirmed by some decrease in the total concentration of paramagnetic species in the sample upon calcination above 600 °C (Table 3). Another reason for the decrease in the linewidth of Mn^2+^ ions can be related to the decrease in the sizes of manganese(II)-containing clusters [49,50]. In the case of Mn-Ce catalysts, an increase in the calcination temperature leads to the increase of Mn^2+^ signal contribution to the total spectrum with a simultaneous decrease in the fraction of Mn^4+^ signal, which represents manganese (IV) ions in a solid solution and/or in the dispersed phase of MnO_x_ [48]. The decrease in the intensity of the manganese (IV) signal can presumably be explained by the agglomeration of MnO_x_ particles formed as a result of the solid solution decomposition and/or partial sintering of more dispersed particles of manganese oxides. All that leads to a strong broadening of Mn^4+^ EPR line up to the complete disappearance of the recorded signal of such particles. Similarly, the proportion of paramagnetic defects (oxygen vacancies, V_O_) signal in the series decreases from 17 to 4% (Table 3). These defects can act as precursors for active oxidation sites, since they are able to adsorb oxygen with the formation of its active forms [47]. Typically, such sites are characterized by lower values of the g-factor and linewidth [47], however, at high concentrations of paramagnetic defects or other sites, an EPR-signal can be broadened with a shift of the peak to a weaker field, since the spin system begins to exhibit ferromagnetic properties [45]. It should be noted that the decrease in the fraction of V_O_ signal with an increase in the calcination temperature of the Mn-Ce catalysts is in good agreement with the decrease in the amount of weakly bound oxygen in these samples, which was determined by TPR-H_2_. 

The Mn-Zr samples demonstrate a significant difference compared to the Mn-Ce systems: in particular, there are no signals of Mn^2+^ ions in the spectra (Figure 7). In this case, signals of Mn^4+^ and paramagnetic defects/oxygen vacancies are observed. The proportion of paramagnetic oxygen vacancies in V_O_/Mn^4+^ ratio increases with an increase in the calcination temperature of the samples (Table 3), which also correlates well with the patterns established by TPR-H_2_ of Mn-Zr catalysts.

The EPR spectra of the Mn-Zr-Ce triple oxide system (Figure 8) are similar to the spectra of Mn-Ce binary catalysts. However, the ratio of narrow signals of divalent and broad signals of tetravalent manganese ions differs (Table 3). The proportion of EPR-detected Mn^4+^ in these samples compared to other signals is significantly higher compared to the double Mn-Ce system. In general, for a triple oxide catalyst, no significant differences are observed depending on the calcination temperature up to 600 °C. At calcination temperature of 700 °C the contribution of manganese (II) signal slightly increases with a simultaneous decrease in the contribution of signals from paramagnetic defects and Mn^4+^ cations.

## 4. Discussion

A comparative study of the calcination temperature effect on the catalytic activity of MnO_x_-CeO_2_, MnO_x_-ZrO_2_ and MnO_x_-ZrO_2_-CeO_2_ oxides in CO oxidation reaction showed that the activities of the MnO_x_-CeO_2_ and MnO_x_-ZrO_2_-CeO_2_ catalysts calcined at 500 °C are close and significantly exceed the activity of the MnO_x_-ZrO_2_ sample, which is in accordance with the literature data [29]. At the same time, an increase in the synthesis temperature of the MnO_x_-CeO_2_ catalyst from 400 to 500 °C affects the lattice parameter of the CeO_2_-based solid solution, in that it indicates the diffusion of manganese from its structure. The revealed changes in structural properties contribute to the increase in catalytic activity. The deactivation of MnO_x_-CeO_2_, observed with a further increase in the calcination temperature, is caused by the intensification of solid solution decomposition, which leads to a significant decrease in the specific surface area and the amount of weakly bound oxygen. On the contrary, the X-ray amorphous state with a high specific surface area in the MnO_x_-ZrO_2_ samples prepared at 400–500 °C has a low catalytic activity. Crystallization of the solid solution phase based on ZrO_2_ and its subsequent decomposition at 650–700 °C cause a sharp increase in activity. Based on the TPR-H_2_ data, it can be argued that such behavior of MnO_x_-ZrO_2_ corresponds to the appearance of fine MnO_x_ particles on the solid solution surface, that increases the content of easily reduced weakly bound oxygen. The decrease in the activity of MnO_x_-ZrO_2_ after calcination at 800 °C is due to the completion of the solid solution decomposition and transformation of the cubic modification of ZrO_2_ into the monoclinic one.

The thermal stability of the MnO_x_-ZrO_2_-CeO_2_ system is in intermediate position between the thermal stability of the binary MnO_x_-CeO_2_ and MnO_x_-ZrO_2_ oxides. The high activity of MnO_x_-ZrO_2_-CeO_2_ observed after calcination at 500 °C is retained up to 650 °C in contrast to MnO_x_-CeO_2_. As in the case of binary oxides, the Mn_y_Zr_x_Ce_1−y_O_2−δ_ solid solution gradually decomposes with the diffusion of manganese cations from its structure and the formation of fine MnO_x_ particles. The shift of the catalytic activity profile towards higher synthesis temperatures is caused by the presence of zirconium, since zirconium ions stabilize the fluorite structure by replacing cerium ions and prevent particle agglomeration, as was mentioned above [22,23]. Indeed, the average CSR size is 50 Å in the temperature range from 400 to 600 °C, while it has a value of 70 Å for MnO_x_-CeO_2_. 

An analysis of the structural properties of MnO_x_-CeO_2_, MnO_x_-ZrO_2_ and MnO_x_-ZrO_2_-CeO_2_ catalysts shows that all three series are characterized by similar changes in the phase composition, depending on the synthesis temperature: an increase in temperature leads to the decomposition of Mn_y_Zr_x_Ce_1−y_O_2−δ_ solid solutions with the formation of Mn_2_O_3_ or Mn_3_O_4_ crystalline oxides and cerium/zirconium oxides. Changes in the lattice parameter of mixed oxide (Figure 2) indicate release of Mn ions from the volume of solid solutions and formations of highly dispersed MnO_x_. The maximum activity for each series is associated with the beginning of solid-solution decomposition. The formation of active phase shifts to the high-temperature region with the addition of Zr to the MnOx-CeO_2_ catalyst due to growth of the temperature of solid solution decomposition. On the other hand, an introduction of Ce into catalyst leads to the formation of crystalline mixed oxide based on the fluorite structure at lower temperature than in the case of MnO_x_-ZrO_2_ system. The latter effect enhances activity of the MnO_x_-ZrO_2_-CeO_2_ catalysts prepared at 400–500 °C.

The correlation of the catalytic activity of the MnO_x_-CeO_2_, MnO_x_-ZrO_2_ and MnO_x_-ZrO_2_-CeO_2_ catalysts with the content of weakly bound oxygen and paramagnetic oxygen vacancies in the sample, as well as the ambiguous interrelation between the catalytic activity and the specific surface area, allow us to claim that the dependences of the activity on the calcination temperature are primarily determined by phase transformations occurring in the studied systems. It can be concluded that the highest value of catalytic activity for each of the considered Mn-containing systems is achieved for the samples containing manganese both in the form of ions in the structure of the corresponding substitutional solid solution and in the form of highly dispersed MnO_x_ particles, formed as a result of manganese segregation on the surface of solid solution during its decomposition. This combination of catalytically active states of manganese in the samples of all studied oxides causes the presence of the highest content of easily reduced reactive oxygen, which was established by TPR-H_2_. In accordance with the EPR data, changes in the catalytic activity of the studied oxides correlate with the content of paramagnetic defects, presumably in the form of oxygen vacancies, which can be located both in the composition of the solid solution and the3 MnO_x_ clusters. In the case of a solid solution, the formation of vacancies can occur due to the diffusion of manganese cations from its structure.

A comparison of MnO_x_-CeO_2_, MnO_x_-ZrO_2_ and MnO_x_-ZrO_2_-CeO_2_ showed that MnO_x_-ZrO_2_-CeO_2_ triple oxide system combines the properties of binary oxides, where the presence of zirconium makes it possible to increase the thermal stability of MnO_x_-CeO_2_ from 500 °C up to 650 °C without significant loss of catalytic activity.

## 5. Conclusions

The influence of the calcination temperature of MnO_x_-CeO_2_, MnO_x_-ZrO_2_ and MnO_x_-ZrO_2_-CeO_2_ catalysts on their structural properties and catalytic activity in the CO oxidation reaction has been studied and compared. The highest catalytic activity of MnO_x_-CeO_2_ is observed for the samples activated at 400–500 °C. In the case of MnO_x_-ZrO_2_, the optimal calcination temperature range is 650–700 °C, and for the Mn-Zr-Ce triple oxide system this interval corresponds to 500–650 °C. It has been established by XRD that the thermal stability of the studied oxides is limited by the onset temperature of decomposition of the corresponding solid solution, where manganese ions diffuse from its structure to the surface in the form of fine MnO_x_ particles. Comparison of the oxides calcined at 500, 600, and 700 °C by TPR-H_2_ and EPR methods showed that the high activity of each of the three studied systems at a specific calcination temperature correlates with the content of easily reduced fine MnO_x_ particles and the presence of paramagnetic oxygen vacancies. The MnO_x_-ZrO_2_-CeO_2_ catalyst showed thermal stability up to 650 °C without significant loss of catalytic activity due to the presence of zirconium in its composition.

## Figures and Tables

**Figure 1 materials-15-07553-f001:**
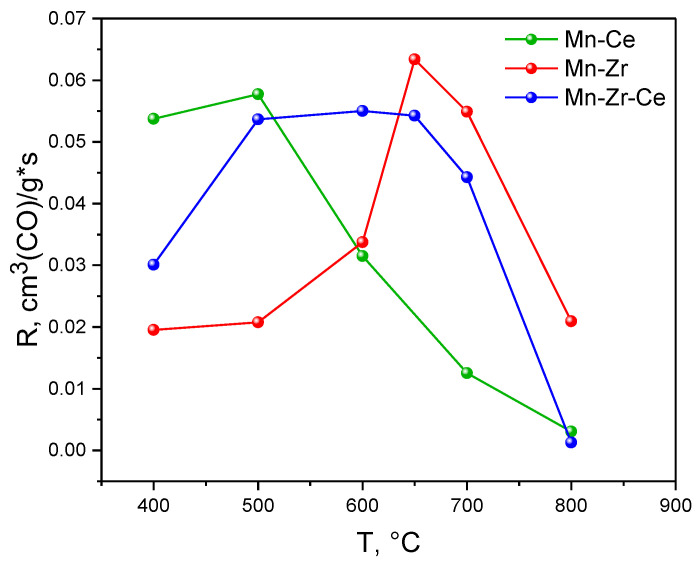
Catalytic activity of Mn-Ce, Mn-Zr and Mn-Zr-Ce samples, calcined at different temperatures, in CO oxidation at 150 °C.

**Figure 2 materials-15-07553-f002:**
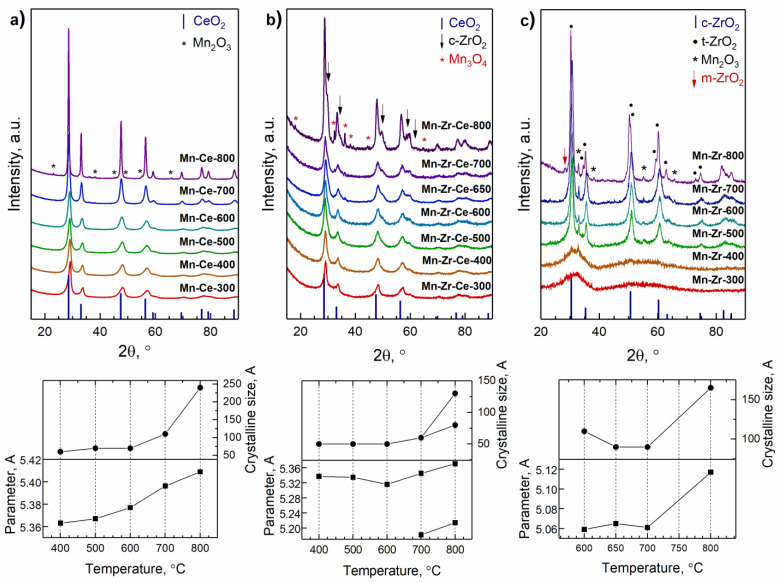
XRD data for (**a**) Mn-Ce, (**b**) Mn-Zr-Ce and (**c**) Mn-Zr catalysts activated at different temperatures.

**Figure 3 materials-15-07553-f003:**
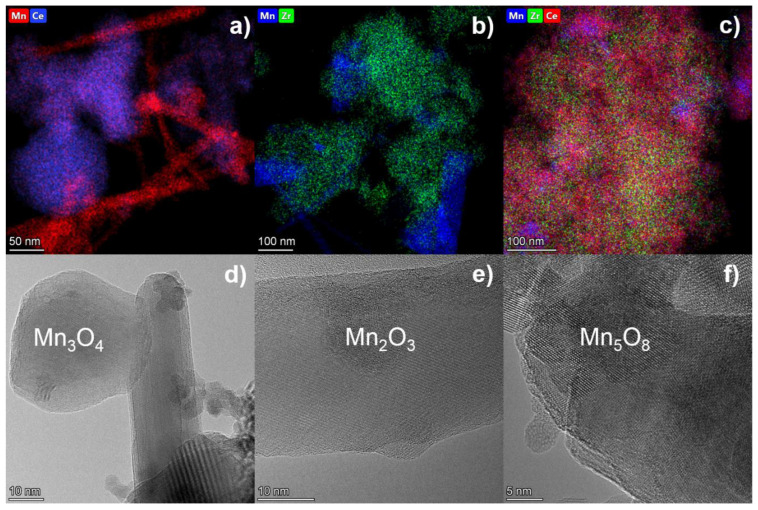
(**a**–**c**) TEM image; (**d**–**f**) EDS mapping pattern for (**a**,**d**) Mn-Ce-500, (**d**,**e**) Mn-Zr-700 and (**c**,**f**) Mn-Zr-Ce-600 catalysts.

**Figure 4 materials-15-07553-f004:**
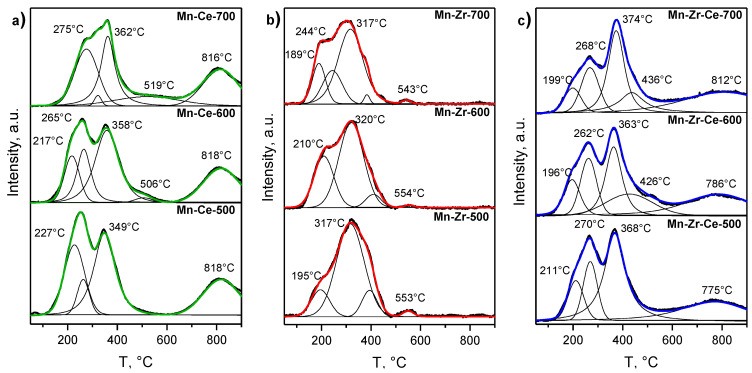
TPR-H_2_ curves for (**a**) Mn-Ce, (**b**) Mn-Zr and (**c**) Mn-Zr-Ce catalysts, calcined at 500, 600 and 700 °C. Diferent colours corresponds to different series.

**Figure 5 materials-15-07553-f005:**
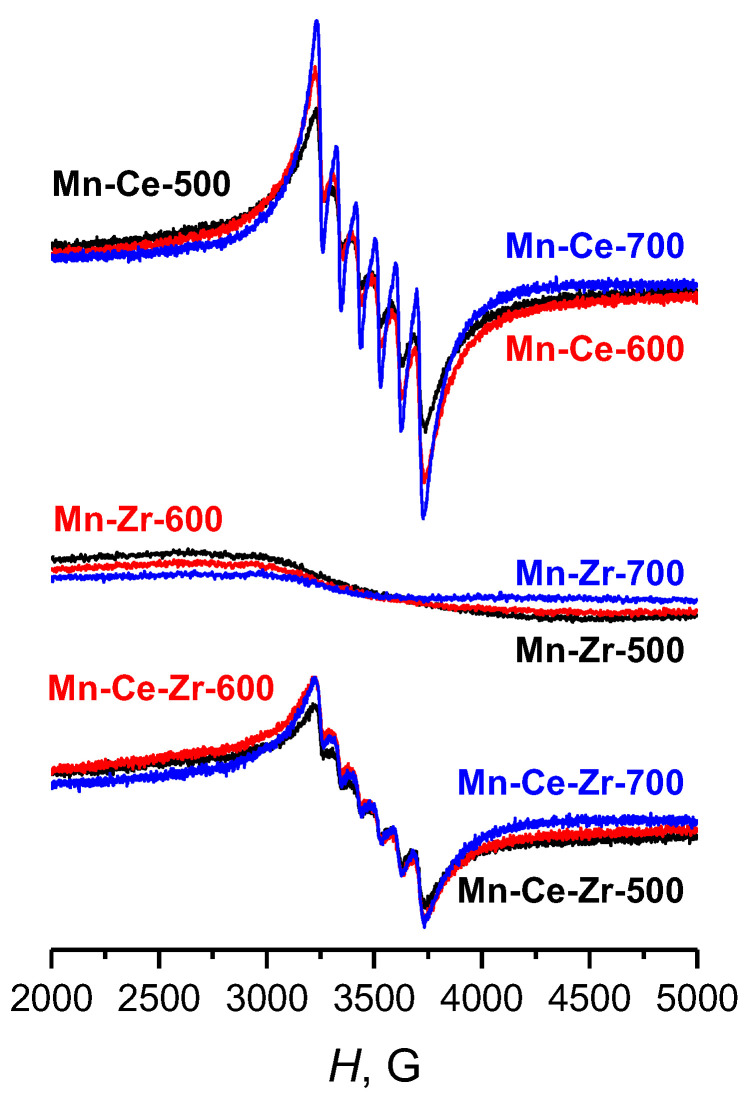
EPR spectra of Mn-Ce, Mn-Zr and Mn-Zr-Ce catalysts, calcined at 500 °C, 600 °C and 700 °C.

**Figure 6 materials-15-07553-f006:**
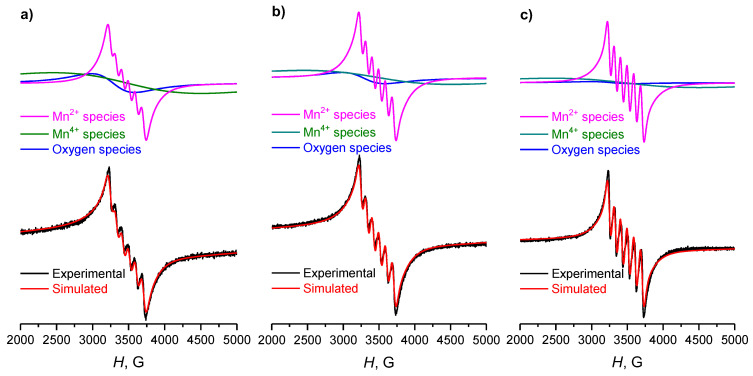
Experimental and simulated EPR spectra of Mn-Ce catalysts, calcined at (**a**) 500 °C, (**b**) 600 °C and (**c**) 700 °C. Different components of the spectra are presented above each simulated curve.

**Figure 7 materials-15-07553-f007:**
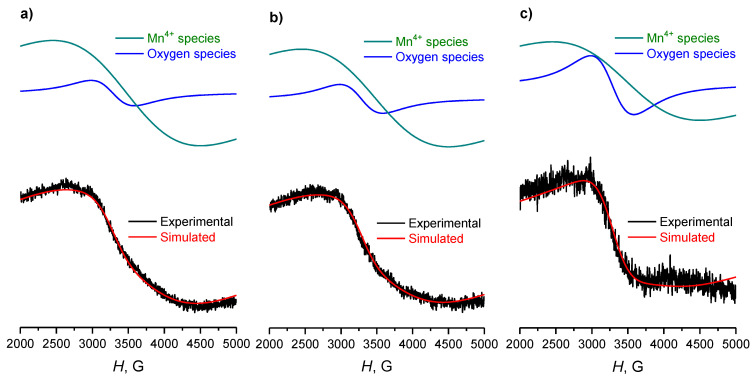
Experimental and simulated EPR spectra of Mn-Zr catalysts, calcined at (**a**) 500 °C, (**b**) 600 °C and (**c**) 700 °C. Different components of the spectra are presented above each simulated curve.

**Figure 8 materials-15-07553-f008:**
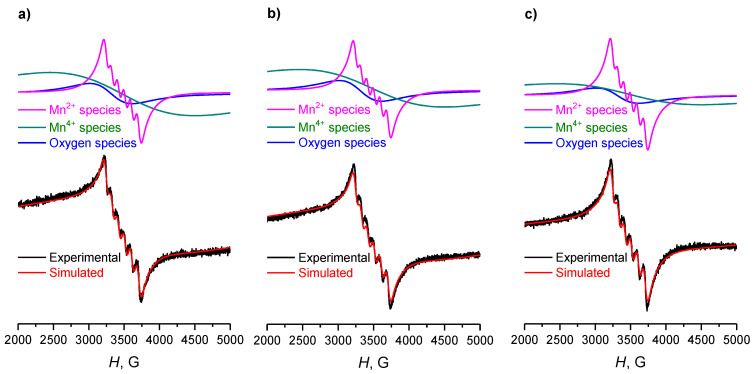
Experimental and simulated EPR spectra of Mn-Ce-Zr catalysts, calcined at (**a**) 500 °C, (**b**) 600 °C and (**c**) 700 °C. Different components of the spectra are presented above each simulated curve.

**Table 1 materials-15-07553-t001:** Structural and microstructural characteristics of Mn-Ce, Mn-Zr and Mn-Zr-Ce samples.

Sample	Phases, wt%	Lattice Constant of Ce(Mn,Zr)O_2_, Å	CSR, Å	S_BET_, m^2^ × g^−1^
Mn-Zr-Ce-400	Ce(Mn,Zr)O_2_ Mn_3_O_4_ (traces)	5.337(1) -	50 130	146
Mn-Zr-Ce-500	Ce(Mn,Zr)O_2_ Mn_3_O_4_ (traces)	5.335(1) -	50 130	110
Mn-Zr-Ce-600	Ce(Mn,Zr)O_2_ Mn_3_O_4_ (traces)	5.316(1) -	50 140	88
Mn-Zr-Ce-650	57% Ce(Mn,Zr)O_2_ 6% Mn_3_O_4_ 37% c-Zr(Ce,Mn)O_2_	5.330(1) 5.169(1)	70 260 60	74
Mn-Zr-Ce-700	55% Ce(Mn,Zr)O_2_ 8% Mn_3_O_4_ 37% Ce_x2_(Mn_y2_,Zr_z2_)O_2_	5.345(1) - 5.182(3)	80 270 70	63
Mn-Zr-Ce-800	44% Ce(Mn,Zr)O_2_ 13% Mn_3_O_4_ 43% Ce_x2_(Mn_y2_,Zr_z2_)O_2_	5.371(1) - 5.214(1)	130 340 80	12
Mn-Zr-400	Amorphous	-	-	304
Mn-Zr-500	Amorphous Mn_2_O_3_	-	-	306
Mn-Zr-600	Amorphous Zr(Mn)O_2_ Mn_2_O_3_	5.059(1) ^1^	- 110 375	176
Mn-Zr-650	98% Zr(Mn)O_2_ 2% Mn_2_O_3_	5.065(1) ^1^	90 410	111
Mn-Zr-700	98% Zr(Mn)O_2_ 2% Mn_2_O_3_	5.061(1) ^1^	90 340	66
Mn-Zr-800	77% t-Zr(Mn)O_2_ 14% m-ZrO_2_ 9% Mn_2_O_3_	5.117(2)^1^	165 195 210	24
Mn-Ce-400	Ce(Mn)O_2_	5.360(1)	60	73
Mn-Ce-500	Ce(Mn)O_2_	5.362(1)	70	57
Mn-Ce-600	Ce(Mn)O_2_	5.377(1)	70	41
Mn-Ce-700	Ce(Mn)O_2_	5.397(1)	110	20
Mn-Ce-800	91% Ce(Mn)O_2_ 2% Mn_3_O_4_ 7% Mn_2_O_3_	5.409(1)	240 - 330	9

^1^ lattice parameter was calculated in cubic approximation or corrected to cubic lattice.

**Table 2 materials-15-07553-t002:** TPR data for Mn-Ce, Mn-Zr and Mn-Zr-Ce catalysts, calcined at 500, 600 and 700 °C.

Catalyst	Total Hydrogen Uptake, mmol (H_2_)×g^−1^	Content of Weakly Bound Oxygen, %
Mn-Ce-500	2.00 × 10^−3^	27
Mn-Ce-600	1.76 × 10^−3^	18
Mn-Ce-700	1.63 × 10^−3^	0
Mn-Zr-500	1.38 × 10^−3^	17
Mn-Zr-600	1.17 × 10^−3^	29
Mn-Zr-700	1.37 × 10^−3^	37
Mn-Zr-Ce-500	2.19 × 10^−3^	12
Mn-Zr-Ce-600	2.17 × 10^−3^	12
Mn-Zr-Ce-700	2.09 × 10^−3^	4

**Table 3 materials-15-07553-t003:** The components of simulated EPR spectra and their contribution to overall spectrum.

Catalyst	The Contribution of the Species to Overall EPR Spectrum, %			Intensity (Double Integral), % rel.
	V_O_ ^1^	Mn^4+ 2^	Mn^2+ 3^	
Mn-Ce-500	17	53	30	100
Mn-Ce-600	12	45	43	105
Mn-Ce-700	4	46	50	75
Mn-Zr-500	8	92	-	80
Mn-Zr-600	10	90	-	60
Mn-Zr-700	20	80	-	30
Mn-Ce-Zr-500	13	71	16	110
Mn-Ce-Zr-600	13	68	19	110
Mn-Ce-Zr-700	12	60	28	75

^1^ V_O_—oxygen vacancies; g = 2.08-2.10, ∆H_pp_ = 600 G. ^2^ g = 1.998, ∆H_pp_ = 2000 G. ^3^ g = 2.00, A = 92 G, ∆H_pp_ = 240 G and ∆H_pp_ = 80 G (Mn-Ce-500), ∆H_pp_ = 65 G (Mn-Ce-600), ∆H_pp_ = 55 G (Mn-Ce-700).

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
