# Peer review of "The Study of Thermal Stability of Mn-Zr-Ce, Mn-Ce and Mn-Zr Oxide Catalysts for CO Oxidation"

_materials, 2022, doi:10.3390/ma15217553_

Round 1

Reviewer 1 Report

 This manuscript studies the effect of the calcination temperature in the range of 400–800°C on the catalytic activity and structural properties of MnOx–ZrO2–CeO2, MnOx– CeO2, and MnOx–ZrO2 catalysts for CO oxidation. The catalysts were characterized by XRD, N2 adsorption, TPR, TEM, and EPR.

 The manuscript is well written and easy to follow, the analysis of the results and the conclusions are correct then I can recommend its publication in Materials.

Author Response

We are thankful to the referee for review our manuscript.

Reviewer 2 Report

In this manuscript, the authors have investigated the catalytic performances of the Mn-Zr-Ce oxides in CO oxidation as well as their structural stability against the thermal and reaction conditions. The conclusion can be supported by a series of the characterization results of XRD, H2-TPR and EPR profiles. I would like to recommend it for publication after several minor issues were solved.

The specific comments list below:

(1) In section of 1. Introduction, it would be better to briefly introduce the background on the CO oxidation reaction over metal oxide-based catalysts.

(2) In Table 1, the mixture phases for Mn-Zr-Ce-700(800) include Ce(Mn,Zr)O2, Mn3O4 and c-Zr(Ce,Mn)O2 (same to Mn-Zr-Ce-650). The title of Figure 2b and 2c should be examined (wrong order).

(3) The fitting analyses of H2-TPR profiles might be unclear. For instance, the reduction peak at 217 (227) oC was obtained on Mn-Ce-600 (500) catalyst, whereas the peak was absent on Mn-Ce-700. Single standard should be referred for peak fitting. Furthermore, some marks of peak position temperatures are wrong in Figure 4, which should be revised properly.

(4) “Ce4+” in 365 line is missing superscript, while “VO” in line 442 is missing subscript.

Author Response

Response to Review

We are thankful to the referee for useful comments. We made revision of the text of our manuscript according to the recommendations.

  1. This manuscript studies the effect of the calcination temperature in the range of 400–800°C on the catalytic activity and structural properties of MnOx–ZrO2–CeO2, MnOx– CeO2, and MnOx–ZrO2 catalysts for CO oxidation. The catalysts were characterized by XRD, N2 adsorption, TPR, TEM, and EPR.

 The manuscript is well written and easy to follow, the analysis of the results and the conclusions are correct then I can recommend its publication in Materials.

  1. In this manuscript, the authors have investigated the catalytic performances of the Mn-Zr-Ce oxides in CO oxidation as well as their structural stability against the thermal and reaction conditions. The conclusion can be supported by a series of the characterization results of XRD, H2-TPR and EPR profiles. I would like to recommend it for publication after several minor issues were solved.

The specific comments list below:

(1) In section of 1. Introduction, it would be better to briefly introduce the background on the CO oxidation reaction over metal oxide-based catalysts.

Answer: The authors agree with the recommendation. Brief information about the CO oxidation over metal oxide-based catalysts has been added to the introduction section.

(2) In Table 1, the mixture phases for Mn-Zr-Ce-700(800) include Ce(Mn,Zr)O2, Mn3O4 and c-Zr(Ce,Mn)O2 (same to Mn-Zr-Ce-650). The title of Figure 2b and 2c should be examined (wrong order).

Answer: The authors agree with the remark. Figure 2 caption has been corrected.

(3) The fitting analyses of H2-TPR profiles might be unclear. For instance, the reduction peak at 217 (227) oC was obtained on Mn-Ce-600 (500) catalyst, whereas the peak was absent on Mn-Ce-700. Single standard should be referred for peak fitting. Furthermore, some marks of peak position temperatures are wrong in Figure 4, which should be revised properly.

Answer: The peak of hydrogen absorption with a maximum at 217-227°C, characteristic of Mn-Ce-600 (500) catalysts, appears due to the reduction of dispersed MnOx particles [1, 2] that form on the surface of the solid solution during its decomposition. In the case of the Mn-Ce sample calcined at 700°Ð¡, this peak is absent due to the crystallization of MnOx species and formation of the crystalline Mn3O4 phase. Differences in the positions of the peak maxima for samples from the same series calcined at different temperatures (500-700°Ð¡) can be associated with different particle sizes or other reasons.

Indeed, a single standard should be used for peak fitting. However, first of all, this is necessary for the study of massive systems. In our case, the catalysts are a mixture of dispersed phases. It is known that the position of the signals of hydrogen uptake by dispersed particles differs significantly from the position of the corresponding signals for bulk systems. Therefore, in our case, it is necessary to have standards of a certain phase composition and dispersion, which is a very difficult task. Moreover, the synthesis conditions of samples, the precursor of the active component, and other factors can also strongly influence the maxima positions in the TPR-H2 profiles for similar samples [3]. Analyzing the TPR-H2 profiles, we focused on the literature data devoted to the study of Mn-Ce, Mn-Zr and Mn-Zr-Ce disperse systems [4-6]. It can be noted that the reduction peaks of manganese oxides have a complex form, but the most intense signals in most cases appear in the temperature ranges of 250–350°C and 350–500°C and are associated with the transformations MnO2/Mn2O3 → Mn3O4 and Mn3O4 → MnO, respectively. The results obtained in this work are in good agreement with the results obtained by other researchers. Similar results were also obtained by Stobbe E.R. and co-authors for individual manganese oxides [7].

We carefully checked the positions of all peaks on the reduction curves, all marks are correct. The profile were fitted with the smallest amount of components, and we were able to obtain good agreement between the model and experimental curves, which is characterized by a high correlation coefficient of 0.98 - 0.99.

[1] Huang, X.; Li, L.; Liu, R.; Li, H.; Lan, L.; Zhou, W. Optimized synthesis routes of Mnox-ZrO2 hybrid catalysts for improved toluene combustion. Catalysts 2021, 11, 1037-1050. doi: 10.3390/catal11091037

[2] Trawczyn ´ski J., Bielaka B., Mis ´ta W. Oxidation of ethanol over supported manganese catalysts—effect of the carrier. Applied Catalysis B: Environmental 2005, 55, 277–285. doi: 10.1016/j.apcatb.2004.09.005

[3] Kapteijn, F., Singoredjo, L., Andreini, A. Moulijn, J.A. Activity and selectivity of pure manganese oxides in the selective catalytic reduction of nitric oxide with ammonia. Applied Catalysis B Environmental 1995, 173-189. doi: 10.1016/0926-3373(93)E0034-9

[4] Huang, X.; Li, L.; Liu, R.; Li, H.; Lan, L.; Zhou, W. Optimized synthesis routes of MnOx-ZrO2 hybrid catalysts for improved toluene combustion. Catalysts 2021, 11, 1037-1050. doi: 10.3390/catal11091037.

[5] Sun, W.; Li, X.; Mu, J.; Fan, S.; Yin, Z.; Wang, X.; Qin, M.; Tadé, M.; Liu, S. Improvement of catalytic activity over Mn-modified CeZrOx catalysts for the selective catalytic reduction of NO with NH3. J. Colloid Interface Sci. 2018, 531, 91–97. doi: 10.1016/j.jcis.2018.07.050.

[6] Azalim, S.; Franco, M.; Brahmi, R.; Giraudon, J.M.; Lamonier, J.F. Removal of oxygenated volatile organic compounds by catalytic oxidation over Zr-Ce-Mn catalysts. J. Hazard. Mater. 2011, 188, 422–427. doi: 10.1016/j.jhazmat.2011.01.135.

[7] Stobbe, E.R., de Boer, B.A., Geus J.W. The reduction and oxidation behaviour of manganese oxides. Catalysis Today. 1999, 47, 161-167. doi: 10.1016/S0920-5861(98)00296-X

(4) “Ce4+” in 365 line is missing superscript, while “VO” in line 442 is missing subscript.

Answer: The authors agree with the remark. The text has been corrected.

Reviewer 3 Report

The paper is very well conceived. The structure and  properties of the materials synthesized are well characterized using several methods generally recommended for this purpose. The results are well presented and discussed. The graphical part is relevant.

The importance of the study is well justified as in the field of catalyst synthesis the calcination temperature may be relevant from efficiency and economical point of view.

The paper may be published in the present form

Author Response

We are thankful to the referees for his work.

Reviewer 4 Report

The influence of the calcination temperature of MnOx-CeO2, MnOx-ZrO2, and MnOx- ZrO2-CeO2 catalysts on their structural properties and catalytic activity for the CO oxidation reaction has been deeply studied in this work. The materials have been characterized with several complementary characterization techniques. The work is very well written and the discussion is based on the experimental results. The topic addressed is of high interest to the field of environmental catalysis and the search for new and resistant materials.
This manuscript can be accepted after the authors comment on a few minor points:

- Could the authors indicate the space velocity or the volume of gas and the mass of catalysts used to obtain the results presented in figure 1?
- The caption of figure 2 should be revised. Figure 2b) corresponds to the Mn-Zr-Ce system.
- Regarding the XRD of the Mn-Zr system, the authors mention that zirconia adopts a cubic structure at a very low temperature (600°C). Were these results supported by any other characterization technique? for example Raman spectroscopy.?On the other hand, are the peaks symmetrical? Could it not be zirconia with a metastable tetragonal structure? Please comment on this and add new citations to clear readers' doubts.
- Could the authors compare the activity of the catalysts presented in this work with that found in similar systems for this reaction?

Author Response

The influence of the calcination temperature of MnOx-CeO2, MnOx-ZrO2, and MnOx- ZrO2-CeO2 catalysts on their structural properties and catalytic activity for the CO oxidation reaction has been deeply studied in this work. The materials have been characterized with several complementary characterization techniques. The work is very well written and the discussion is based on the experimental results. The topic addressed is of high interest to the field of environmental catalysis and the search for new and resistant materials.
This manuscript can be accepted after the authors comment on a few minor points:
- Could the authors indicate the space velocity or the volume of gas and the mass of catalysts used to obtain the results presented in figure 1?

Answer: Each point in Figure 1 is the result of averaging activity values at three different contact times (gas flow rates were 253, 370 and 487 ml/min). The obtained activity values were close. This confirms the reliable determination of activity in the kinetic reaction region, since a change in the contact time leads to a proportional change in CO conversion: as the flow rate increases, the conversion decreases. According to the activity calculation equation (given in the experimental part), it can be seen that changes in the gas flow rate and CO conversion compensate each other, and the activity should remain constant. In this case, the mass of the catalyst was not constant and was chosen so that the conversion did not exceed ~30% in order to avoid the appearance of diffusion restrictions.

- The caption of figure 2 should be revised. Figure 2b) corresponds to the Mn-Zr-Ce system.

Answer: The authors agree with the remark. Figure 2 caption has been corrected.
- Regarding the XRD of the Mn-Zr system, the authors mention that zirconia adopts a cubic structure at a very low temperature (600°C). Were these results supported by any other characterization technique? for example Raman spectroscopy.?On the other hand, are the peaks symmetrical? Could it not be zirconia with a metastable tetragonal structure? Please comment on this and add new citations to clear readers' doubts.

Answer: We agree with reviewer. From XRD data for low temperature zirconia, it is not possible to distinguish the cubic and tetragonal. Although this modification of ZrO2 can not be considered strictly cubic, its lattice parameter was calculated in the cubic approximation because of the low calcination temperature and because of the small size of the obtained particles that led to the broadening of diffraction peaks. The sentence «The diffraction pattern of this sample also contains peaks at 2θ = 30.5, 35.4, 50.9, 60.6, 63.6, 74.9, 83.0, and 85.7°, which correspond to the 111, 002, 022, 113, 222, 004, 133, 024 reflections of ZrO2 oxide in the cubic modification (space group Fm3m, PDF â„– 77-2157).» was replaced by «The diffraction pattern of this sample also contains peaks at 2θ = 30.5, 35.4, 50.9, 60.6, 63.6, 74.9, 83.0, and 85.7°, which correspond to the ZrO2 oxide. Since the broadening of diffraction peaks, from XRD data it is not possible to correct distinguish the cubic and tetragonal modification of zirconia.» Also we have changed the information in Table 1.

- Could the authors compare the activity of the catalysts presented in this work with that found in similar systems for this reaction?

Answer: Despite the fact that similar catalysts have been extensively studied in the oxidation reactions of various substrates, it is problematic to make the required comparison. The catalytic properties of MnOx-CeO2(ZrO2) are often presented in the literature as a dependence of «Substrate Conversion vs. Reaction Temperature». Unfortunately, we did not find information about the catalytic properties of MnOx-CeO2(ZrO2) samples in the CO oxidation reaction in the form of catalytic activity at 150°C.

Reviewer 5 Report

At present, studies of various multi-cation oxide systems with specific compositional, lattice, and charge characteristics are attracting much attention; their synthesis, crystallization, and valence charge density distribution features are also of particular interest. The research work presented in this paper is of broad interest; the topic is important for studies in materials science and solid-state chemistry because the stoichiometry, characteristics and specific properties caused by mixed-valence states closely correlate with the chemical composition, internal microstructure, band structure, size and morphology of multi-cation compounds. The paper is well organized and properly referenced. The analysis is consistent and supported by experimental work. Thus, the manuscript is worthy of publication.

There are several aspects, summarized below, that the authors might consider:

1) The style of the manuscript needs some improvement. The title is unclear. The description of the purpose and objectives of the paper is limited. A more extended version of the conclusion regarding the structure-property relationships could also enhance the manuscript. Several phrases in the text seem unclear in a crystal-chemical or crystallographic sense. For example, "the distribution of electronic states of manganese", "lattice oxygen", "the role of the cationic environment on the manganese cations state and the change in catalytically active states of the samples", "space group Fm3m", etc. (there is such space group name "Fm-3m").

2) Page 6, Figures 2b and 2c : Could one suggest that the characteristic feature (the minimum) in the behavior of the lattice parameter shown in the bottom parts of both figures indicates some phase transformation (transition)?

Author Response

At present, studies of various multi-cation oxide systems with specific compositional, lattice, and charge characteristics are attracting much attention; their synthesis, crystallization, and valence charge density distribution features are also of particular interest. The research work presented in this paper is of broad interest; the topic is important for studies in materials science and solid-state chemistry because the stoichiometry, characteristics and specific properties caused by mixed-valence states closely correlate with the chemical composition, internal microstructure, band structure, size and morphology of multi-cation compounds. The paper is well organized and properly referenced. The analysis is consistent and supported by experimental work. Thus, the manuscript is worthy of publication.

There are several aspects, summarized below, that the authors might consider:

1) The style of the manuscript needs some improvement. The title is unclear. The description of the purpose and objectives of the paper is limited. A more extended version of the conclusion regarding the structure-property relationships could also enhance the manuscript. Several phrases in the text seem unclear in a crystal-chemical or crystallographic sense. For example, "the distribution of electronic states of manganese", "lattice oxygen", "the role of the cationic environment on the manganese cations state and the change in catalytically active states of the samples", "space group Fm3m", etc. (there is such space group name "Fm-3m").

Answer:

According to reviewer recommendation, we have extended the purpose and objectives, change the article title. The information of structure-property relationships in the manuscript was added into the manuscript (Introduction and Discussion part).

Page 2. «Comparison of three series of catalysts would help to determine structure – activity relationship, including the role of the cationic composition in formation of solid solution, temperature of decomposition, the change in the catalytically active states.»

Page 15-16. «An analysis of structure properties for MnOx-CeO2, MnOx-ZrO2 and MnOx-ZrO2-CeO2 catalysts shown that all three series are characterized by similar changes in the phase composition depending on the synthesis temperature: an increase in temperature leads to the decomposition of MnyZrxCe1-yO2-δ solid solutions with the formation of Mn2O3 or Mn3O4 crystalline oxides and cerium/zirconium oxides. Changes in the lattice parameter of mixed oxide (Figure 2) indicates release of Mn ions from the volume of solid solutions and formations of highly disperse MnOx. The maximum activity for each series is associated with the begging of solid solution decomposition. The formation of active phase shifts to the high-temperature region with the addition of Zr to the MnOx-CeO2 catalyst due to grow of the temperature of solid solution decomposition. On the other hand, an introduction of Ce into catalyst leads to the formation of crystalline mixed oxide based on the fluorite structure at lower temperature than in the case of MnOx-ZrO2 system. The latter effect enhances activity of the MnOx-ZrO2-CeO2 catalysts prepared at 400-500°C.»

The phrases «the distribution of electronic states of manganese», «lattice oxygen», «the role of the cationic environment on the manganese cations state and the change in catalytically active states of the samples», «space group Fm3m» were replaced by «the states of manganese ions», «mobile oxygen» / «oxygen», «Comparison of three series of catalysts would help to determine structure – activity relationship, including the role of the cationic composition in formation of solid solution, temperature of decomposition, the change in the catalytically active states.» « space group Fm-3m» consequently.

2) Page 6, Figures 2b and 2c : Could one suggest that the characteristic feature (the minimum) in the behavior of the lattice parameter shown in the bottom parts of both figures indicates some phase transformation (transition)?

Figure 2 shows the dependence of the mixed oxide lattice parameter on the calcination temperature. An increase in the calcination temperature leads to a sharper change in the lattice parameters of the solid solutions formed in the Mn-Ce-Zr and Mn-Zr samples (at 700°C) compared to the solid solution existing in the Mn-Ce system, which is characterized by a linear change in the lattice constant. In the case of Mn-Zr and Mn-Ce-Zr catalysts at 400-700°C, there is a little variation of lattice parameters. The origin of such variation may be the phase transformation (from cubic to tetragonal modification), change the oxidation state of Mn ions (Mn4+/Mn2+ ions go to Mn2+), combination of an exit of the Mn ions from bulk of mixed oxide and change the oxidation state, segregation of Mn ions in the volume of mixed oxide and formation of two types of mixed oxides. From our results, it is not possible to correct distinguish such factors. In any case, in this process Zr ions play an important role.